# The Multiple Facets and Disorders of B Cell Functions in Hepatitis B Virus Infection

**DOI:** 10.3390/jcm12052000

**Published:** 2023-03-02

**Authors:** Dilhumare Ablikim, Xiaoqing Zeng, Chunli Xu, Mengxiao Zhao, Xuecheng Yang, Xuemei Feng, Jia Liu

**Affiliations:** 1Department of Infectious Diseases, Union Hospital, Tongji Medical College, Huazhong University of Science and Technology, Wuhan 430022, China; 2Joint International Laboratory of Infection and Immunity, Huazhong University of Science and Technology, Wuhan 430022, China; 3Department of Anesthesiology, Union Hospital, Tongji Medical College, Huazhong University of Science and Technology, Wuhan 430022, China

**Keywords:** hepatitis B virus, B cell, antibody, adaptive immune response

## Abstract

Chronic hepatitis B virus (HBV) infection continues to be a global public health burden. B cells play a pivotal role in mediating HBV clearance and can participate in the development of anti-HBV adaptive immune responses through multiple mechanisms, such as antibody production, antigen presentation, and immune regulation. However, B cell phenotypic and functional disorders are frequently observed during chronic HBV infection, suggesting the necessity of targeting the disordered anti-HBV B cell responses to design and test new immune therapeutic strategies for the treatment of chronic HBV infection. In this review, we provide a comprehensive summary of the multiple roles of B cells in mediating HBV clearance and pathogenesis as well as the latest developments in understanding the immune dysfunction of B cells in chronic HBV infection. Additionally, we discuss novel immune therapeutic strategies that aim to enhance anti-HBV B cell responses for curing chronic HBV infection.

## 1. Introduction

According to the World Health Organization, nearly a third of the world’s population has been infected with the hepatitis B virus (HBV) at some point in their lives [1]. Most adults who are exposed to HBV develop a self-limiting infection and successfully control the virus, with less than 5% of adults developing HBV persistence [1,2,3]. In contrast, about 90% of these who acquire HBV perinatally or in early childhood develop chronic infections. Despite the availability of highly effective prophylactic vaccines against HBV, over 257 million people worldwide have been affected with chronic hepatitis B (CHB), which leads to severe sequelae, including liver fibrosis, cirrhosis, and hepatocellular carcinoma [4,5].

Currently, there are two types of drugs approved as first-line therapies for CHB in clinical guidelines: nucleot(s)ide analogs (NUCs) and pegylated interferon alpha (PEG-IFNα) [6,7]. Although current therapies can efficiently control HBV infection, they rarely lead to a functional curing of CHB. which means achieving persistently undetectable serum HBV surface antigen (HBsAg), with or without the appearance of hepatitis B surface antibody (anti-HBs) [8]. This is mainly because HBV covalently closed circular DNA (cccDNA) minichromosomes and HBV DNA are incorporated into the host genome and are thus difficult to eliminate using these drugs [9,10,11]. 

The outcome of HBV infection is mainly determined by virus–host immune system interactions, and host adaptive immune responses play a key role in mediating HBV clearance [12]. In recent decades, most research focuses on anti-HBV adaptive immunity, such as in T cell immunity or T cell exhaustion in chronic HBV infection, whereas B cell responses have long been neglected [13]. As one of the two key arms of adaptive immunity, the B-cell-mediated humoral immune response leads to the generation of HBV-specific antibodies, including anti-HBs and antibodies to the HBV e (anti-HBe) and core (anti-HBc) antigens, which correlate with immune protection and the transition between the clinical phases of HBV infection [14]. Moreover, B cells also participate in the development and regulation of anti-HBV cellular immunity through various mechanisms, such as antigen presentation and immune regulation (Table 1). Recent advancements in generating fluorescently labeled HBsAg and HBV core antigen (HBcAg) baits have enabled the quantification and functional characterization of HBV-specific B cells in an unprecedented manner, which has greatly improved our understanding of B cell immune dysfunction in CHB [15] (Table 2). Here, we reviewed the multiple roles of B cells during HBV infection at both the phenotypic and broader functional levels and summarized the current promising CHB treatment strategies targeting anti-HBV B cell responses. 

## 2. Multiple Roles of B Cells in HBV Infection

### 2.1. Antibody Production

Early knowledge of the roles of B cells in HBV infection was primarily obtained via the detection of serum antibodies against different HBV component proteins, which have important clinical implications for diagnosing the disease, distinguishing clinical phases, and evaluating disease severity [49,50]. Detection of anti-HBs is associated with protective immunity against HBV, indicating recovery from an acute hepatitis B virus infection or immunity from HBV vaccination [20]. HBV is a hepatotropic virus that specifically infects hepatocytes by binding to functional receptors on the membrane, such as heparan sulfate proteoglycans (HSPGs) [51,52] and sodium taurocholate co-transporting polypeptide (NTCP) [53,54]. Anti-HBs can potently inhibit HBsAg binding to HSPGs and thus have a strong neutralizing activity [55]. Antibodies against a highly conserved motif in the pre-S1 domain of HBsAg exert a strong additional neutralizing activity by inhibiting the interaction between the pre-S1 region and NTCP, thereby blocking HBV entry [16,56,57]. Anti-HBs immune complexes (HBsAg-IC) could be detected in the majority of patients with CHB. In those who achieved functional cure (FC), HBsAg-IC peaks overlapped with alanine aminotransferase (ALT) flares. In contrast, overlapping HBsAg-IC and ALT peaks were not detected in non-FC patients. Moreover, HBsAg epitope diversity was stable in FC patients, while it decreased in non-FC patients within the first 12–24 weeks of NUC treatment [17]. A high anti-HBs titer could significantly improve the sustained functional cure rate and may be clinically beneficial for patients with CHB undergoing PEG-IFN-based therapy [58,59]. 

The Fab and Fc domains of antibodies act together to drive antibody effector function. The Fab domain specifically binds to antigens, while the Fc domain can recruit complement and innate immune cells to engage in effector functions, including complement dependent cytotoxicity (CDC), antibody-dependent cellular cytotoxicity (ADCC), and antibody-dependent cellular phagocytosis (ADCP), etc. Anti-HBs can initiate the formation of HBsAg-IC, which can then be cleared by intrahepatic phagocytes such as Kupffer cells and circulating phagocytic cells, including monocytes, B cells, dendritic cells, and neutrophils through ADCP [18]. It is also believed that HBsAg-IC may recruit NK cells into the liver and increase apoptosis of HBV-infected hepatocytes through ADCC during HBV infection [19,60]. In contrast to the protective role of anti-HBs, the generation of anti-HBc has been demonstrated to participate in mediating liver injury during HBV infection through CDC. In HBV-associated acute liver failure patients, the HBcAg attached on the surface of hepatocytes can be recognized by high-affinity intrahepatic anti-HBc, leading to widespread CDC and liver damage [21,22]. 

Detection of anti-HBc suggests HBV infection. Anti-HBc IgM is a serum marker of acute HBV infection or an indication of serious aggravation of chronic infection. Anti-HBc IgG is found in patients throughout the history of HBV infection, from ongoing to resolved HBV infection phases, and even in occult HBV infection [20,61]. Anti-HBc primes immune complex formation by activating the classical complement pathway, mediates complement-dependent cytotoxicity, and increases HBV-infected hepatocyte lysis [19]. Moreover, anti-HBc levels in serum may reflect the degree of hepatic inflammation and thus be used as an indicator for antiviral treatment decisions in patients with CHB with normal ALT levels [62,63]. Baseline anti-HBc levels also strongly predict HBeAg seroconversion in patients with CHB treated with Peg-IFN or NUC therapy [64]. 

HBeAg seroconversion, which means the loss of serum HBeAg and the detection of anti-HBe antibodies, could spontaneously occur in the natural history of CHB [65]. Although anti-HBe antibodies cannot neutralize HBV virions, their presence is considered an indicator of the establishment of partial immune control of HBV replication. This is mostly associated with the decrease of severity of liver disease of the patients and transition to the inactive carrier state. High levels of anti-HBe often predict better disease outcomes [23,24].

### 2.2. Antigen Presentation and T/B Cell Interaction 

Virus-specific T-cell responses are fundamental for the successful control of HBV infection [66]. The generation of effective anti-HBV T-cell responses needs the participation of antigen-presenting cells (APCs). Aside from generating antibodies, B cells are also classified as professional APCs like DCs and macrophages, as they possess all the mechanisms needed for antigen uptake, processing, and presentation, including major histocompatibility complex (MHC) I and MHC II, costimulatory molecules CD80 and CD86, etc. [67]. Antigen-specific B cells are extremely efficient in ingesting cognate antigens through their high-affinity B cell receptors (BCRs) [68,69]. Ingested antigen polypeptides are processed and loaded onto MHC molecules and then transported to the cell surface for presentation [70,71]. HBcAg-specific B cells have been shown to bind and internalize HBcAg with high efficiency through specific membrane-surface immunoglobulin receptors and present antigens to CD4+ T cells far more efficiently than macrophages or DCs [40,72]. Notably, aside from MHC II molecules, B cells express relatively high levels of MHC I molecules [25]. It has been demonstrated that splenic B cells separated from HBV protein-vaccinated mice could also cross-present HBcAg to specific CD8+ T cells via MHC I and induce HBcAg-specific cytotoxic T lymphocyte (CTL) response [26,27]. Similarly, HBsAg-specific B cells can efficiently cross-present HBsAg to CD8+ T cells and induce HBsAg-specific CTL response [28,29]. However, the induced CTLs in turn lead to HBsAg-specific B cell killing, and thus suppress the production of anti-HBs antibodies by B cells. This mechanism is believed to promote HBV persistence during infection [30]. 

Consistent with these earlier studies, a recent study characterized the transcriptome features of circulating HBcAg- and HBsAg-specific B cells from patients with CHB and demonstrated that the two HBV-specific B cell populations share similar mRNA expression patterns, which show increased expression of genes linked to antigen cross-presentation (XCL1 and CD40LG) and innate immune activation (MYD88, IFNA1/13, IFNA2, IFNB1) [15]. The role of B cells as APCs in HBV infection has also been supported by increasing clinical observations of HBV reactivation following treatment with rituximab, an anti-CD20 antibody that is generally used to exhaust B cells during chemotherapy in patients with B cell lymphoma [73,74,75]. The population at risk for HBV reactivation includes not only those who are currently infected with HBV, but also those who have resolved HBV infection [76]. The risk of HBV reactivation differs according to the host factors, virologic factors, and the type and degree of immunosuppression. The host factors associated with an increased risk of reactivation include male sex, older age, and cirrhosis development [77], and the virologic factors include higher baseline HBV DNA and anti-HBc levels and HBsAg and HBeAg positivity [78,79]. Among patients with a resolved HBV infection, the presence of serum anti-HBs seems to be a protective factor against HBV reactivation [73]. HBV reactivation can occur in a variety of settings and typically is associated with immunosuppressive therapies. Of all immunosuppressive therapies, the highest HBV reactivation rates are reported following B-cell-depleting treatments [77].

Except as antigen presentation, T/B cell interactions are also important for B cell activation and functional regulation. Follicular helper T (Tfh) cells are a special CXCR5 expressing CD4^+^ T cell subset that directly interact with B cells in germinal centers (GCs) and help them to produce antibodies [31]. The interaction between Tfh cells and B cells promotes both the ability of B cells to differentiate into high-affinity long-lived plasma cells and memory B cells and the formation of GCs [32]. An early study proved that HBcAg-specific Tfh cells can help circulating B cells produce antibodies against not only HBcAg, but also HBsAg [33,80]. HBsAg is strictly a T cell-dependent antigen; it has been shown that Tfh cells are required for anti-HBs production and HBsAg seroconversion [81]. Consistent with these findings, a study using an HBV infection mouse model and investigating patients with CHB showed that the Tfh cell response to HBsAg is essential for HBV clearance [82]. Clinically, the frequency of circulating Tfh cells was discovered to be significantly increased in patients with CHB compared to healthy controls and is also associated with disease progression, suggesting that a high frequency of CXCR5+ CD4+ Tfh cells are a likely a biomarker for assessing the immune status of patients with CHB [40]. Oral application of TLR8 agonist in CHB patients could induce IL-21 production by Tfh cells and enhance circulating HBsAg-specific B cell responses [33]. Moreover, CXCR5 expression is also found on a subset of CD8+ T cells in GCs [34]. This unique group of cells express costimulatory molecules, CD40L and ICOS, that are included after activation; these molecules are also involved in humoral immune responses [34]. During chronic HBV infection, this unique group of CXCR5+ CD8+ T cells show a less exhausted status and possess stronger antiviral abilities than CXCR5- CD8+ T cells [34,35]. 

### 2.3. Immune Regulation 

Upon activation, B cells can produce different cytokine profiles that regulate immune responses and thus can be divided into two major subsets: effector B (Beff) cells and regulatory B (Breg) cells [83]. Beff cells produce proinflammatory cytokines such as IL-6, IFN-γ, and TNF-α to promote proinflammatory immune responses and memory CD4+ T cell responses [36,37]. Moreover, these cytokines may limit HBV replication through their direct antiviral activities, such as inducing degradation of HBV cccDNA, reducing HBV transcription, and inhibiting HBV entry by regulating NTCP expression in hepatocytes [84]. In contrast, Breg cells produce IL-10 and TGF-β suppressive cytokines [85] and play an important role in suppressing effector T cell function and enhancing regulatory T (Treg) cell function to induce HBV immune tolerance [38,39]. Studies have shown that IL-10-producing Breg cells can inhibit HBV-specific CD8+ T cell responses, and depletion of those cells may restore specific CD8+ T cell responses [80,86,87]. Moreover, Breg cells suppress the antiviral effector cytokine production of CD4 + T cells, downregulate the proinflammatory Th17 response, and promote the generation of CD4 + CD25 + Treg cells, which suppress anti-HBV CTL responses [84,88].

Myeloid-derived suppressor cells (MDSCs) are a specialized immunosuppressive cell population derived from myeloid progenitor cells, which expand during infection, cancer, and inflammation and control the functions of other immune cells [89]. The expansion of MDSCs during HBV infection has been frequently observed in HBV animal models and CHB patients [90,91,92] and is believed to be induced by HBeAg stimulation [38]. It has been reported that MDSCs separated from PBMCs of CHB patients in different phases of infection, including the immune-tolerant phase and HBeAg+/HBeAg- chronic hepatitis B phases, could induce the expansion of Tregs through producing TGF-β and IL-10 [85]. MDCSs have also been shown to reduce the proliferation of mature human B cells and suppress their homing and antibody production by secreting soluble factors, such as NO/ROS, PGE2, and TGF-β [39]. Moreover, MDSCs could induce the expansion of Bregs via iNOS and enhance their immunosuppressive functions [93]. It has been shown that Bregs can educate both the monocytic and granulocytic MDSCs of mice and humans by stimulating their immunosuppressive properties [94]; therefore, there may exist a reciprocal regulation between MDSCs and Bregs during HBV infection.

## 3. Immune Dysfunction of B Cells during CHB Infection

### 3.1. Phenotypic Changes of B Cells during CHB Infection 

HBV infection has a notable impact on the phenotypes of both global and HBV-specific B cells [48]. Early studies demonstrated that circulating B cells in patients with CHB showed an hyperactivated phenotype compared to healthy controls; the activation marker of CD69 was increased along with higher expression levels of transferrin receptor CD71, Toll-like receptor (TLR) 9, and liver-homing marker CXCR3 [40,41]. Analysis of global circulating B cells of CHB patients in the immune active and inactive carrier phases by RNA sequencing revealed that the transcriptome of these cells showed an activated status; CD83, CD300c, CXCR4, and CD69 levels were upregulated, and a panel of innate-stimulating genes were also increased [42]. In line with these observations, Salimzadeh et al. also reported an activating feature of upregulated of CD83 expression in sorted memory B cell (MBC) subsets from patients with CHB [43]. Additionally, the gene expression profiles of intrahepatic B cells differed from patients with CHB-paired peripheral B cells, showing an upregulated BCR expression and activation of multiple immune signaling pathways, suggesting the presence of extrafollicular MBC formation [95,96]. 

Description of the HBV-specific B cell phenotype is technically difficult; however, recent technical developments in generating fluorescently labeled HBsAg and HBcAg baits have allowed for the characterization and quantification of the functionality of HBV-specific B cells. Salimzadeh et al. and Burton et al. demonstrated that HBsAg-specific B cells appear at a similar frequency in acute, chronic, and resolved HBV infections, and both circulating and intrahepatic HBsAg-specific B cells isolated from patients with acute and chronic HBV infection failed to mature into anti-HBs-secreting cells in vitro [43,45]. These cells resemble atypical memory B cells (atMBC) characterized by low CD21 and CD27 expression and high expression of inhibitory markers, such as PD-1 and the transcription factor T-bet [44]. HBsAg-specific atMBC clusters have been detected in liver infections and inflammation sites [95,96]. Notably, higher frequencies of circulating HBcAg-specific B cells could be detected than HBsAg-specific B cells in the same patient with CHB at different disease stages. Moreover, HBcAg-specific MBCs show a higher CD95 expression and lower IL-10Ra expression compared with HBsAg-specific B cells, suggesting that they are phenotypically more activated. Most HBcAg-specific B cells have the IgG+ memory B cell phenotype and could mature in vitro into antibody-secreting cells [15,44]. The differences in the phenotypes and functions of HBsAg and HBcAg-specific B cells in the same patient suggest that a high HBsAg level might cause the dysfunctional programming of HBsAg-specific B cells through persistent stimulation [15,97]. 

Costimulatory molecules are important cell surface molecules for the antigen presentation function, such as CD80, CD86, and CD40. Thus, analysis of their expression on B cells may help evaluate the antigen presentation function of the cells [98,99]. The expression of CD86 on circulating B cells showed no significant change in HBeAg+ CHB patients [100]. In contrast, the expression of CD80 on circulating B cells was found to be reduced in CHB patients at phases of immune tolerance, immune activation, and immune clearance, and this decrease could be inverted after the HBV infection subsided [58,101]. Costimulatory molecules of CD40 expression on B cells were also observed to be decreased in CHB infection [102], indicating a potentially impaired antigen presentation function of B cells during chronic HBV infection.

### 3.2. Immune Dysfunction of B Cells during CHB Infection 

Although the activated B cells of patients with CHB displayed significantly lower proliferative ability compared to those of healthy donors, the ability of global B cells in peripheral blood mononuclear cells (PBMCs) to produce IgG in vitro was preserved [58,101]. This is supported by clinical observations that the antibody responses induced by the hepatitis A virus (HAV) or COVID-19 vaccine in patients with CHB are comparable to those in healthy controls (HCs) [103,104]. However, numerous studies have confirmed that PBMCs isolated from patients with CHB are defective in producing anti-HBs antibodies [105]. Consistent with these observations, recent studies have shown that circulating HBsAg-specific B cells isolated from patients with both acute and chronic HBV infection show compromised ability in vitro to proliferate and mature into antibody-secreting cells [43,45,48]. Blocking immune inhibitory signaling pathway PD-1/PD-L1 or providing addition costimulatory signals such as IL-2, IL-21, and CD40L may partly regain the function of HBsAg-specific B cells [43,45,48]. Interestingly, these studies also suggest that defects in HBsAg-specific B cells in producing anti-HBs antibodies are formed early after HBV exposure and remain so, unrelated to the outcome of the infection [46]. In contrast to HBsAg-specific B cells, the antibody-producing function of HBcAg-specific B cells seems to be intact during CHB infection, and the portion of HBcAg-specific B cells with a CD27- CD21- AtMBC profile is markedly lower than that of HBsAg-specific B cells [45]. Furthermore, the number of circulating HBcAg-specific MBCs and the ability of PBMCs to produce anti-HBc antibodies were correlated with ALT levels in patients with CHB. HBcAg-specific MBCs showed a higher activated memory phenotype in disease phases with higher ALT levels than those with low ALT levels [15,46,47], suggesting a correlation between HBcAg-specific humoral immunity and the development of natural clinical phases of CHB. Overall, these studies suggest that the antibody-producing function of HBsAg-specific B cells is defective but partially rescued, while HBcAg-specific B cells are abundant, and their activation is associated with disease progression in patients with CHB. 

A disorder in the immune regulatory function of B cells has also been observed during CHB infection. Compared with patients with acute hepatitis B (AHB) and HCs, the concentration of IL-10 in the supernatant and the percentages of Breg and Treg cells of PBMCs in patients with CHB were significantly increased [86]. IL-10 may inhibit the CD8+ T cell response to proinflammatory cytokine secretion by enhancing regulatory T cell function [40]. In patients with CHB and CHC, the number of circulating IL-10+ Breg and Tfr cells was positively correlated with serum ALT and AST levels, and the number of circulating IL-10+ Breg cells and Tfr/Tfh ratio in patients with HBeAg+ CHB was markedly higher than those in patients with HBeAg- CHB [106].

## 4. Therapeutic Strategies for CHB Infection Targeting B Cell Responses

The current approved first-line therapies for CHB treatment in clinical guidelines include NUCs and PEG-IFNα, which rarely promote HBsAg seroconversion. However, influenced B cell homeostasis can be partially recovered through these antiviral treatments [107], suggesting that the recovery of the functional impairment of B cells is possible during CHB infection. The restoration of the anti-HBV B cell response may contribute to HBsAg seroconversion in patients with CHB through different mechanisms: (i) anti-HBs antibodies can bind to intrahepatic virions, preventing infection of new hepatocytes or binding to HBsAg on the liver cell membrane to promote ADCC [108]; (ii) a high HBsAg load could inhibit adaptive immune function and eventually lead to HBV tolerance. Thus, recovery of B cell function can decrease high HBsAg loads and further promote anti-HBV T cell responses to eradicate HBV infection [13,81].

In the last two decades, many attempts targeting HBV-specific B-cell responses have been made to restore and enhance them by using conventional or modified HBV vaccines, known as “therapeutic vaccination” [109]. Traditional prophylactic vaccines with yeast-derived HBsAg have a limited effect in inducing HBsAg clearance in patients with CHB. Therefore, for therapeutic purposes, new-generation vaccines with enhanced immunogenicity are required to induce anti-HBs more efficiently and achieve a functional cure for CHB [110,111,112]. It has been shown that clinical patients with CHB present lower immune tolerance to the preS1 domain of the HBV large surface antigen than the small HBsAg; therefore, vaccines targeting the preS1 region may improve therapeutic effects in chronic HBV infection. In line with this hypothesis, Yingjie et al. showed that HBV carrier mice showed a less tolerized status to HBsAg after vaccination with the preS1-polypeptide, which allowed these mice to respond to the HBsAg vaccine. Thus, sequential administration of antigenically distinct preS1-polypeptide and HBsAg vaccines in HBV carrier mice successfully induced anti-HBs seroconversion and the resolution of chronic HBV infection [113]. In another pre-clinical study performed in rhesus macaques, novel DNA vaccine adjuvants consisting the S, PreS1, and Core antigens were shown to induce specific immune responses, and anti-PreS1 antibodies were induced more rapidly than anti-S or anti-Core antibodies after vaccination [114]. A clinical trial in patients with CHB also demonstrated that repeated monthly injections of a recombinant Pre-S1/Pre-S2/S HBV vaccine co-administered with daily oral lamivudine treatment could suppress HBV replication and lead to anti-HBs seroconversion in about half of the treated patients [115]. Recently, Zhang et al. used a novel immuno-enhanced virus-like particle carrier (CR-T3) derived from the roundleaf bat HBV core antigen to display a unique B-cell epitope, SEQ13 (HBsAg-aa113-135), forming the candidate molecule CR-T3-SEQ13. A sustained decrease of HBsAg and HBV DNA levels could be induced in HBV transgenic mice after CR-T3-SEQ13-based vaccination. Moreover, CR-T3-SEQ13-based vaccination led to a complete resolution of intrahepatic HBV replication in hydrodynamic injection-based HBV carrier mice. The antiviral effect of CR-T3-SEQ13 vaccination therapy was strongly correlated with anti-HBs levels, suggesting that the vaccine-induced SEQ13-specific antibody response is the major factor that mediates HBV clearance [116].

In addition to therapeutic vaccination, another strategy is to enhance B cell activation and anti-HBV function by providing costimulatory signals, such as stimulation with TLR agonists or costimulatory ligands. TLR ligands act as adjuvants to stimulate primitive B cells to induce antibody production and promote B cell differentiation into long-term viable plasma cells [117,118,119]. The ability of oral TLR8 agonists to promote Tfh and B-cell responses was tested in phase 1b clinical trials, and TLR8 agonists enhanced HBV-specific B-cell responses in patients with CHB by improving the monocyte-mediated Tfh function [33]. A recent study also demonstrated that treatment with the oral TLR7 agonist JNJ-64794964 could induce sustained plasma HBV-DNA and HBsAg loss in a chronic HBV infection mouse model, and no rebound of viremia was observed 4 weeks after halting the JNJ-4964 treatment. High serum anti-HBs levels and HBsAg-specific IgG-producing B cells, and IFNγ-producing CD4+ T cells were detected in the spleens of JNJ-64794964-treated mice [120]. CD40 activation substantially improves antigen presentation function of normal and malignant B cells, and CD40-activated B cells efficiently induce naïve and memory CD4+ and CD8+ T-cell responses [121]. Liu et al. initially demonstrated that B cells from patients with CHB could be activated by soluble CD40 ligands, and activated B cells loaded with the HBcAg polypeptide could induce the generation of HBV-specific T cells [119,122]. Moreover, costimulatory cytokine IL-12-based vaccine therapy has also been shown to restore systemic HBV-specific CD4+ T cell responses, eliciting robust hepatic HBV-specific CD8+ T cell response, and facilitate the generation of HBsAg-specific humoral immunity, which in turn could reverse liver-induced immune tolerance toward HBV [123,124]. 

## 5. Conclusions

B-cell functionality during HBV infection is multi-dimensional. B cell functions, including antibody secretion, antigen presentation, and immune regulation, play important roles not only in mediating HBV clearance but also in the pathogenesis and persistence of HBV infection. The recent development of HBV bait-staining techniques, combined with advanced single-cell methods, has provided unprecedented insights into the phenotypic and functional dysfunction of HBV-specific B cells and can drive further research on the detailed mechanisms behind B cell dysfunction during CHB infection. Novel CHB treatment strategies targeting B cells can potentially be developed based on an advanced understanding of these mechanisms. The restoration of antiviral B cell function and the development of desired anti-HBV B cell responses may finally help us to achieve a functional cure for CHB.

## Figures and Tables

**Table 1 jcm-12-02000-t001:** Multiple roles of B cells in hepatitis B virus infection.

B Cell Functions	Classification	Roles in HBV Infection	Refs.
Antibody production	Anti-HBs	(1) Block HBV entry and replication (2) Induce immune complex formation to promote HBV clearance through ADCC, ADCP, and CDC (3) Serum marker of resolution of HBV infection or gaining immunity via HBV vaccination	[16,17,18,19]
	Anti-HBc	(1) Prime immune complex formation, activate the classical complement pathway by mediating CDC, and increase HBV-infected hepatocyte lysis (2) IgM: Serum marker of acute HBV infection; indication of serious aggravation of chronic infection (3) IgG: Serum marker of prior or ongoing HBV infection	[19,20,21,22]
	Anti-HBe	Indicate the establishment of partial protective immunity against HBV and a favorable outcome	[23,24]
Antigen presentation	HBcAg-specific B cells	(1) Bind and internalize HBcAg with high efficiency and present them to CD4+ T cells (2) Induce an HBcAg-specific CTL response	[25,26,27]
	HBsAg-specific B cells	(1) Efficiently cross-present serum HBsAg to CD8+ T cells (2) Induce HBsAg-specific CTL response, but may in turn lead to killing of HBsAb-producing B cells	[28,29,30]
T/B cell interaction	CXCR5+CD4+ Tfh cells	Help B cell maturation and promote antibody production in germinal centers	[31,32]
	CXCR5+CD8+ T cells	Control HBV replication during chronic hepatitis B	[33,34,35]
Immune regulation	Effector B cells	Produce proinflammatory cytokines to facilitate proinflammatory immune response and memory CD4+ T cell responses	[36,37]
	Regulatory B cells	Produce suppressive cytokines to inhibit effector T cells and enhance regulatory T cell function	[38,39]

ADCC: antibody-dependent cell-mediated cytotoxicity; ADCP: antibody-dependent cellular phagocytosis; CDC: complement-dependent cytotoxicity; CTL: cytotoxic T lymphocyte.

**Table 2 jcm-12-02000-t002:** Immune dysfunction of B cells during chronic hepatitis B infection.

B Cell Dysfunctions	Classification	Subtype	Characteristics	Refs.
Phenotypic changes during CHB	Global B cells	Circulating B cells	(1) Higher expression of CD69, CD71, and TLR-9 in CHB (2) Upregulated CD83, CD300c, CXCR4, and CD69 in the immune active and inactive carrier phases	[40,41,42]
		Memory B cells	Upregulated CD83 expression	[43]
	HBV-specific B cells	HBcAg-specific B cells	(1) Can efficiently mature into antibody-secreting cells in vitro (2) Nearly all have the IgG+ memory B cell phenotype	[15,44]
		HBsAg-specific B cells	(1) Fail to mature into anti-HBs-secreting cells in vitro (2) Resemble atypical memory B cells (atMBC)	[43,44,45]
Immune dysfunction during CHB	HBV-specific B cells	HBcAg-specific B cells	HBcAg-specific B cells are abundant, and their activation is associated with disease progression in patients with CHB.	[15,46,47]
		HBsAg-specific B cells	Could not expand and mature efficiently into antibody-secreting cells in vitro	[43,45,48]

## Data Availability

No new data were created or analyzed in this study. Data sharing is not applicable to this article.

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
