# Peer review of "The Multiple Facets and Disorders of B Cell Functions in Hepatitis B Virus Infection"

_jcm, 2023, doi:10.3390/jcm12052000_

Round 1
Reviewer 1 Report
In their manuscript, the authors make an overview of the knowledge about B cells biology in the case of HBV infection. HBV interaction with the immune system is of particular interest to understand pathogenesis and develop innovative cures. B cells are important to control viral infection because of their specific feature as detailed by the authors.
I would have few questions and remarks concerning the manuscript.
In general, I would recommend to the authors to precise B cells origin when it is possible (liver or circulating blood). Phenotype could be different from the circulating blood than in the liver, site of HBV infection.
Patient status in CHB is not often mentioned and B cells phenotypes described could be different according to the phase of the CHB ( HBe+,HBe-...)
B reg cells and T reg cells are mentioned to be part of the chronic disease progression but it would be important to include the MDSC.
Is the high level of HBsAg the only explanation about B cells dysfunction ? Is there any scale of HBsAg quantity leading to a stronger B cell dysfunction ?
HBsAg-specific B cells are described here to be not affected as the HBsAg-specific. However, antibodies against HBcAg in CHB are often produced later than the HBeAg antibodies in the natural disease evolution. What can explain this delayed production if specific-B cells are fully functional ?
As a last notice I would recommend a careful proofreading of the manuscript to correct English mistakes.
Reviewer 2 Report
In this study entitled “The Multiple Facets and Disorders of B Cell Functions in Hepatitis B Virus Infection”. The authors provide a review of the characteristics of Hepatitis B infection alongside a comprehensive immunological summary of B cells roles in CHB. Although this type of work is remarkable in infectious disease and clinical virology fields, it is crucial that the information and evidence be well structured and well-written.
Here are some minor points to consider if possible:
1- I suggest including further details on the role of immunosuppression therapy in the context of HBV infection, as its contributing to worsening prognosis.
2- It would be helpful to clarify further the roles of ADCP, ADCC, and CDC, as well as their crucial impact on HBV clearance.
